# Polymeric Carriers Designed for Encapsulation of Essential Oils with Biological Activity

**DOI:** 10.3390/pharmaceutics13050631

**Published:** 2021-04-28

**Authors:** Aurica P. Chiriac, Alina G. Rusu, Loredana E. Nita, Vlad M. Chiriac, Iordana Neamtu, Alina Sandu

**Affiliations:** 1Department of Natural Polymers, Bioactive and Biocompatible Materials, Petru Poni Institute of Macromolecular Chemistry, 700487 Iasi, Romania; rusu.alina@icmpp.ro (A.G.R.); lnazare@icmpp.ro (L.E.N.); neamtui@icmpp.ro (I.N.); sandu.alina@icmpp.ro (A.S.); 2Faculty of Electronics Telecommunications and Information Technology, Gh. Asachi Technical University, 700050 Iași, Romania; vchiriac@etti.tuiasi.ro

**Keywords:** essential oil, natural and synthetic polymeric carriers, biological activity, machine learning

## Abstract

The article reviews the possibilities of encapsulating essential oils EOs, due to their multiple benefits, controlled release, and in order to protect them from environmental conditions. Thus, we present the natural polymers and the synthetic macromolecular chains that are commonly used as networks for embedding EOs, owing to their biodegradability and biocompatibility, interdependent encapsulation methods, and potential applicability of bioactive blend structures. The possibilities of using artificial intelligence to evaluate the bioactivity of EOs—in direct correlation with their chemical constitutions and structures, in order to avoid complex laboratory analyses, to save money and time, and to enhance the final consistency of the products—are also presented.

## 1. Introduction

The interest for the current use of essential oils (EOs) is confirmed by the need to address the issue in specific fields through compounds and natural structures, through the large number of dedicated scientific articles, and is supported by the EOs market that is significantly driven by the growing demand for natural products, which usually have less or no alteration, at the same time with raising awareness of health and well-being among consumers [1]. The rising of the consumer awareness about the use of EOs is mainly due to their medical and relaxing properties, which are the major driving factors of the market [2].

The use of EOs in medicine, pharmaceutical sciences, biology, and agronomy dates back to ancient time due to their specific bioactive molecules, composed essentially from flavonoids, isoflavones, aldehydes, terpenoids, phenolic acids, carotenoids, and alkaloids, which provide specific activities, such as antioxidative, anti-inflammatory, cytoprotective, antitumor, anthelmintic, antimicrobial, antihypertensive, analgesic, larvicidal, insecticidal, antiparasitic, and other biological activities. EOs are suitable as complementary medicinal treatments, owing to their pleasantness and availability, and also, to synergistic therapeutic effects with the prescribed medical treatments, e.g., antibiotics [3,4,5]. Aspects related to the significant antimicrobial potential against multidrug-resistant pathogens of the EOs, their synergetic effect when more oils are mixed [6,7,8], or synergistic activity when used in combination with known drugs [9,10,11,12,13,14], are also presented in a recent review [15]. The reviews are generally intended to present an overview of the antimicrobial properties of EOs, their mechanisms of action, components of EOs, nano-encapsulated EOs, and synergistic combinations of EOs, in order to apply them and overcome multidrug-resistant microorganisms [16,17]. They Reviews are also devoted to the elucidation of the mechanisms of action of the oils, the effects of EOs on different microorganisms, or how they work in combination with other antimicrobial compounds.

A particularly important direction is represented by the problem of EO encapsulation, as well as exploiting the synergism between oil constituents and medicines to produce cooperation and/or a combined effect.

Another systematic analysis of the current research on lemongrass, lavender, clove, dill, and other EOs related to their anti-convulsing activities, which can be beneficial for people with epilepsy, as recently reviewed by Bahr et al. [18]. The authors mentioned the main constituents responsible for these beneficial effects, such as as asarone, carvone, citral, eugenol, and linalool compounds, while the internal use of sage, hyssop, rosemary, camphor, pennyroyal, eucalyptus, cedar, thuja, and fennel oil have effects on epileptic seizures due to the presence of thujone, 1,8-cineole, camphor, or pino-camphone, which have been identified as convulsive agents. Concerning the mechanisms of action, the authors attributed (to oils) the ability to modulate the GABAergic system of neurotransmission, and alter the ionic currents through ion channels.

Vaillancourt et al. investigated the synergistic interactions between thyme and winter savory oils, with penicillin G, chlorhexidine, or nisin against penicillin-resistant *S. hyicus* 84-2978 (thyme and winter savory) and *S. aureus* 25923 (cinnamon, thyme, and winter savory) [19]. The study provided evidence that EOs, namely cinnamon, thyme, and winter savory, may contribute toward developing herbal treatments against exudative epidermitis in piglets. As the researchers presented the therapeutic potential of EOs, used as topical therapeutic agents against exudative epidermitis, may reduce the incidence of the disease and decrease the utilization of antibiotics to treat this infection.

The review by Magdalena Valdivieso-Ugarte et al. underlined the use of EOs in clinical studies, including: (i) efficacy against foodborne pathogens, namely the inhibition of *S. aureus*, *V. cholerae*, and *C. Albicans*; (ii) antioxidant activities in a dose range of 0.01 to 10 mg/mL in cell models, owing to the presence of the phenolic compounds; and (iii) immune-modulatory activities attributed to their ability to modify the secretion of cytokines [20]. The authors evidenced the benefits conferred by using EOs, as well as the need to continue investigations to confirm their biological properties, and support activities of EOs for normalization of dose and incubation times in cell and animal models.

EOs, as secondary metabolites with a wide range of bioactivities, especially antimicrobial properties, are utilized to treat various human ailments and diseases. The capacity of EOs for cancer cell targeting activity, conferring the ability to increase the efficacy of commonly used chemotherapy drugs (e.g., paclitaxel and docetaxel), and pro-immune functions when administered to cancer patients, was presented in the reviews [21]. The articles are intended to be state-of-the-art research in regards to (1) the application of EOs as anticancer agents, both in vitro and in vivo, and (2) their use in combination with conventional chemotherapeutic strategies, targeting cancer cell specificity.

Based on clinical scientific evidence in favor of EO use in various phases of pre- and postoperative treatments—including the mechanisms of action of inhaled aromatherapy, starting with the absorption of volatile molecules and their transformations into chemical signals to produce characteristic physiological and psychological effects—Susanna Stea et al. elaborated (in a review) on the use of EOs as complementary treatments for surgical patients [22]. The review includes state-of-the-art research in the field of EOs.

Natural compounds of EOs have emerged as potential candidates in dentistry—being developed as preventive or therapeutic agents for various oral diseases, given the biotechnological focus in the search for antimicrobial and dental therapy [23]. The article provides information on the possibility of EOs contributing toward improving the quality of dental treatments.

Numerous articles have reviewed the potential of using EOs due to their multiple and various phyto-constituents, that stand out for the complexity of oils compositions, as well as by their large number of biological activities and effects able to generate [24]. The article by Sara García-Salinas et al. for example, presented an evaluation of the antimicrobial activities and cytotoxicity of the different components (of natural origin) found in EOs [25]. The authors evidenced the highest in vitro antimicrobial activities against *Escherichia coli* and *Staphylococcus aureus* (*S. aureus*) of carvacrol, cinnamaldehyde, and thymol, developed through a bactericidal mechanism identified by membrane disruption. EO presence in concentrations above 0.5 mg/mL hinders *S. aureus* biofilm formation and elimination of the preformed biofilms. The sub-cytotoxic values (0.015–0.090 mg/mL) are lower than minimum inhibitory and bactericidal concentrations for bacteria, but much higher than chlorhexidine doses (0.004 mg/mL). At the same time, the minimum bactericidal concentrations would only be three orders of magnitude higher than the sub-cytotoxic dose.

In a recent review, the use of approximately 90 EOs with at least 1500 combinations for dermatological infections was presented [26]. Their antimicrobial properties were found against pathogens, such as *S. aureus*, *Streptococcus pyogenes*, *Propionibacterium acnes*, *Haemophilus influenzae*, and *Brevibacterium* species. The review revealed the possibilities for EOs encapsulation in composites and blends, as well their interactions with conventional antimicrobials for synergic effects. The challenges in this context include finding solutions (i.e., the combination of EOs, oils, and drugs), as well as finding suitable carriers for these compounds.

Other EOs, for example *Lippia origanoides*, *Thymus vulgaris*, *Lippia alba*, *Cymbopogon martini*, *Cymbopogon flexuosus*, *Rosmarinus officinalis*, *Salvia officinalis*, *Swinglea glutinosa*, *Tagetes lucida*, *Satureja viminea*, *Cananga odorata*, *Citrus sinensis*, and *Elettaria cardamomum*, were investigated for their antimicrobial, antibiofilm, and anti-Quorum sensing (QS) activities against *Escherichia coli* and *Staphylococcus epidermidis* [27,28,29,30].

The antibiofilm activities of some EOs, namely eugenol, carvacrol, or copaiba oil [31,32], against a large spectrum of pathogenic bacterial strains, and their isolated components, were recently investigated and reviewed [33]. The review by Barros presented a suggestive illustration of the mechanisms, by which the nano-formulations and nanoparticles conjugated with natural products could affect bacterial biofilms (Figure 1).

Other authors highlighted the natural products as interesting sources for quorum sensing inhibitors (QSIs), due to their vast chemical diversities with structural complexities and biological activities, offering a promising tool in the treatment of bacterial infections, including those that are biofilm-related [34,35,36,37].

A recent review recommended the nanoparticles (NPs) associated with natural products and phytochemicals as promising platforms for antibiofilm technologies, in connection with studies about antibiofilm compounds, and in relation to the necessity of obtaining these new materials [38]. The article justified applicability of NPs due to the antimicrobial activities of EOs, of their extracts or isolated compounds, when they can be found on the surfaces of NPs, encapsulated, or combined in formulations, such as nanoemulsions. In this context, the ability of oils to produce microemulsions and nanoemulsions that show availability alongside antimicrobial activities of being drug delivery vehicles should also be mentioned.

Franklyne, in a review, presents the potential of EO as a drug delivery vehicle for water insoluble drugs and drugs with low intestinal absorption [39]. The mechanism of the antimicrobial actions of EOs, which cause damage to the bacterial cell wall or cell membranes, inducing complete cell disorganization and disruption, is also mentioned. Moreover, it is also highlighted that the presence of peripheral cytoplasmic condensations [40], irreversible loss of viability [41], increase in cell membrane permeability [42], and partial solubilization of the cell membrane by fatty acids, leads to the release of membrane proteins and uncoupling of oxidative phosphorylation, or enzyme and nutrient uptake inhibition; thus, causing bacteriostasis [43].

**Figure 1 pharmaceutics-13-00631-f001:**
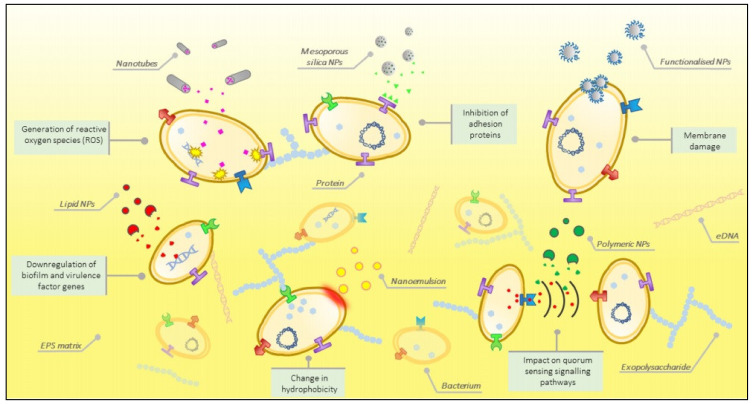
Illustration of mechanisms that can affect bacterial biofilms by nano-formulations and nanoparticles conjugated with natural products. Reprinted with permission from ref. [38]. Copyright 2020, American Chemical Society.

Orchard et al.—concerning EO use against dermatological infections—presented the coupled effect of the carrier oils on antimicrobial activities and cytotoxicity of EOs against skin pathogens [44]. EOs are rarely utilized undiluted, and the use of an oil carrier as a base for blending is made before applying the EO to the skin to prevent irritation (e.g., contact dermatitis). In Orchard’s article, the positive effects of the oil carrier over cytotoxicity, without antagonizing the EO’s antimicrobial activity, was presented. Thus, the study identified *Aloe vera* and *Schisandra chinensis* as oils carriers that caused the highest reduction of cytotoxicity and increased antimicrobial activity, while *Aloe vera*, *Schisandra chinensis, Hypericum perforatum, Calendula officinalis*, and *Persea americana* were found to increase the antimicrobial activity of several EOs against respective pathogens, including *Brevibacterium epidermidis*, *Brevibacterium. linens*, and *Pseudomonas aeruginosa*.

In the review by Tomasz M. Karpiński, the antifungal activities of EOs from 72 Lamiaceae plants were presented [45]. The author presented the EOs as alternatives to medicines due to their broad antimicrobial spectrums, in the context of increased incidences of fungal infections, from recent years, and the high frequency of infections caused by drug-resistant strains and new pathogens, e.g., *Candida auris*. The best activity (minimum inhibitory concentrations, MICs < 100) of EOs from some species of the genera *Clinopodium*, *Lavandula*, *Mentha*, *Thymbra*, and *Thymus* was also mentioned. The chemical components most commonly found, as the main ingredients, were β-caryophyllene (41 plants), linalool (27 plants), limonene (26), β-pinene (25), 1,8-cineole (22), carvacrol (21), α-pinene (21), p-cymene (20), γ-terpinene (20), and thymol (20).

There is a need to discover the maximum and prolonged beneficial effects of EOs in the various fields in which these products find applicability. As a result, numerous studies research the protective coverings of EOs and their complex preparations to successfully manifest their biological properties.

The encapsulation of bioactive oils through various methods (e.g., settled or used interdependently with EO systems), were realized due to their low water solubility, strong organoleptic flavor, and low stability, but also for their protection from oxygen, light, moisture, and heat, and the intended usages. Rodriguez, in his review, presented current encapsulation strategies for bioactive oils (Figure 2) [46].

Thus, the encapsulation of EOs into polymer-based formulations, for example, micelles, micro- and nanocapsules, films, or hydrogels, is provided to ensure decreasing volatility, improved stability, and water solubility. In this way, the properties of these active ingredients are maintained unaltered or even increased in effectiveness, and a control over the release of the encapsulated molecules can be provided when necessary, as mentioned in the review by Bilia [47].

EOs encapsulation also prevent the oxidative deterioration and loss of volatile compounds, in order to maintain their biological activities (e.g., antibacterial, antiviral, antifungal, anticancer, antidiabetic, anti-inflammatory, antioxidant, antiprotozoal, and insect-repellent), and their compositions after exposure to light, moisture, oxygen, and high temperature, and for easier handling and industrial processing [48].

Studies on the potential of EO encapsulation in various synthetic or natural polymer systems, to improve human health and wellbeing, and found in the current literature, are presented in this context.

Polymeric nanoencapsulation of EOs through nanoprecipitation techniques recently received tremendous attention. The method is a well-established approach that ensures improved water solubility, effective protection against degradation, prevention of volatile components evaporation, and controlled and targeted release. These benefits were underlined in the review by Lammari [49]. In the article, the method and applicability of the nanotechnology were correlated with the chemical composition of EOs, the principle of polymeric nanoparticle preparation, the physicochemical properties of EOs loaded nanoparticles in relation to their current applications.

The cyclodextrin properties, namely the internal hydrophobic cavity and the external hydrophilicity, make it one of the best compounds used for the protection of products with poor solubility, such as EOs [50]. Thus, the complexation of EOs in cyclodextrins ensures their stability, as well as the subsequent controlled release of these compounds. As a result, this procedure is a solution utilized for the protection of many phytochemicals, but the use of cyclodextrins for EO encapsulation is not the purpose of this review. Cyclodextrins are included only due to their multiple uses and importance.

Along with the polymeric matrices utilized for encapsulating EOs, these structures can provide multiple benefits. The literature mentions potential in corroborating the activities between EOs and metal/metal oxide nanocomposites, in conferring a broad spectrum of bioactivities, including antioxidant, anticancer, and antimicrobial attributes, as presented in the article by Basavegowda [51].

## 2. Polymeric Carriers for EOs Encapsulation

### 2.1. Natural Polymers

The use of natural macromolecular compounds for EO encapsulation represents the first option, due to their biocompatibility, biodegradability, and the provenience from renewable natural resources, in the context of increasing concern of undesirable environmental and socioeconomic consequences.

As mentioned previously, the interest in using EOs against various diseases has increased, mainly because oil composition can be correctly and deeply analyzed; thus, one can certainly appreciate the effect generated by these compounds after application. Their use was also made for neglected diseases caused by various classes of pathogens, including protozoa, viruses, bacteria, and helminths, and for which the interest of pharmaceutical companies in healing was, in some cases, non-existent, as presented in the review by Luna [52]. The following are some of the most commonly used natural polymer matrices for loading with EOs.

#### 2.1.1. Alginate

Alginate is generally utilized due to its chemical stability, pH sensitivity, capacity to form strong gel barriers against water and gases, and biological functionality in appetite regulation [53,54]. Mina Volić et al. realized a study for the preparation of a carrier of thyme oil, which is known for its expectorant, antitussive, anti-broncholytic, antispasmodic, antihelminthic, carminative, muscle relaxant and diuretic properties [55]. The polymer–thyme oil system, based on the mixture of alginate and soy protein isolate (SPI), was made by emulsifying thyme oil in an aqueous sodium alginate–SPI blend solution, followed by atomization via electrostatic extrusion, and then crosslinking with Ca^2+^ ions. The composite was chosen to combine the pH sensitivity of alginate with the bioactivity and emulsifying properties of SPI. The new system was capable of intestinal delivery of the EO. The formation of a complex microsphere with a thyme encapsulation efficiency of 72–80% was confirmed after determination of the total polyphenols content, and an oil release of about 42–55%.

SPI, a partner of alginate, in the previous reference, was investigated for nanocomposites film preparation with sodium montmorillonite clays that incorporated eugenol to obtain antimicrobial composite films [56]. The study evidenced the interactions between eugenol and clay with the modification of film properties, for example, the increase of the glass transition temperature of the films. The retention of eugenol in SPI films was favored by the clay presence, while the release was slowed.

Eugenol, a hydroxyphenyl propene natural compound with several pharmacological activities, was presented in the review by Barboza et al. from the viewpoint of its anti-inflammatory and antioxidant properties, mechanisms of action, and therapeutic potential for the treatment of inflammatory diseases [57]. Eugenol, which exerts beneficial effects on human health, found applicability in numerous bioactive structures. Its excellent antimicrobial behavior—activity against fungi and a wide range of Gram-negative and Gram-positive bacteria, targeting different kinds of microorganisms, such as those responsible for human infectious diseases, diseases of the oral cavity, and foodborne pathogens—were reviewed and analyzed [58].

The alginates are used as matrices for oil encapsulation through technologies based on the external, internal, or inverse gelation mechanisms, due to their high gelling capacity, biocompatibility, low toxicity, and capability of forming three-dimensional networks in the presence of calcium ions [59,60,61,62]. The review by Martins [59] presented the oil incorporation techniques using alginate as an encapsulating agent, including the dispersion-crosslinking and dispersion–coacervation techniques.

A more applied article refers to the ability to encapsulate EOs in alginate beads, by using the electrostatic extrusion technique, while retaining their antioxidant potential [63]. The study evaluated the chemical composition and antioxidant potential of lavender (*Lavandula angustifolia*), tea tree (*Melaleuca alternifolia*), bergamot (*Citrus bergamia*) and peppermint (*Mentha piperita*) EOs by the 2,2-diphenyl-1-picrylhydrazyl (DPPH), and 2,2′-azinobis(3-ethylbenzothiazoline-6-sulfonic acid) (ABTS) methods, and their antimicrobial activities. It was revealed that lavender and bergamot EOs were more efficient in inhibiting bacterial growth than other tested oils, with the minimum inhibitory concentration of 5 μg/mL.

An alginate/cashew gum system was prepared via spray drying for developing biopolymer nanoparticles blends for encapsulation of EOs [64]. The study utilized alginate as a polyanionic linear biopolymer consisting of β-D-manuronic acid and α-L-guluronic acid, which was able to interact with divalent cations crosslinking agents to form the so-called “egg-box” complexes for coupling various EOs [65], while cashew gum was used as a drug carrier, gelling agent, and colloidal stabilizer [66,67,68,69]. In the presented procedure, in the solution containing alginate and cashew gum, Lippia sidoides oil and Tween 80 as an emulsifier were slowly added. The prepared emulsion was crosslinked with CaCl_2_ solution under mechanical stirring at 18,000 rpm, and then spray dried. The characterized nanoparticles presented an average size in the range of 223–399 nm, and zeta potential values ranged from −30 to −36 mV, attesting excellent stability. EO content varied from 1.9 to 4.4% with an encapsulation efficiency of up to 55%. The in vitro release profile of EO was between 45 and 95% within 30–50 h, exhibiting activity against the larvae of the dengue vector *Aedes aegypti*.

The use of alginate macromolecular chains crosslinked with calcium ions (Ca^2+^) for the treatment of highly exuding wounds and burns is well known [70,71]. In recent study, researchers investigated the possibility of developing a multifunctional dressing, capable of both protecting the wound from external agents, and promoting the regeneration of new tissue [72]. In order to obtain bioactive nanofibrous dressings by the electrospinning procedure, the researchers combined two naturally derived compounds, namely sodium alginate and lavender EO. The solution was made from sodium alginate, polyethylene oxide, Pluronic F127, and lavender oil to obtain an emulsion ready for the electrospinning process. The in vitro and in vivo investigations performed on the obtained nanofibrous dressings proved their antibacterial activity against *S. aureus*, and inhibition of the pro-inflammatory cytokines production. According to the authors, the prepared electrospun dressings proved efficient in promoting burn healing; they recommended these new dressings as wound care systems.

Another research study investigated the encapsulation of green tea flavan-3-ols and caffeine in large and hard particles, prepared from alginate with soy or hemp protein systems, and in spherical and softer particles based on whey proteins and bovine serum albumin [73]. The highest content of polyphenols and caffeine was retained by the combinations of the alginate and calcium caseinate or whey proteins. The release studies, in water and simulated gastric/intestinal fluids, revealed burst release of polyphenols in the first 5–10 min followed by sustained release up to 120 min.

Electrospinning, which is a versatile technique used for fabrication of nanostructures [74], tissue engineering [75], wound dressing [76], enzyme immobilization [77], electrode materials [78], and food packaging preparation [79], was recently reviewed as a method for EO encapsulation in membranes. The systems prepared by this procedure exhibited fibrous morphology with a large surface area to volume ratio, high porosity, and fiber diameters, in the range of nano to micron, and with favorable properties for the sustained release of active ingredients from the packaging membrane to various surfaces [80].

A natural/synthetic polymer system based on sodium alginate and poly(vinyl alcohol) (PVA) incorporated with cinnamon, clove, or lavender oils was tested by the electrospinning procedure for the manufacture of antibacterial nanoscale fibers [81]. The study was performed, as both polymers have excellent biocompatible properties, while the investigated EOs have the desired antibacterial properties. The research study confirmed the increase of the polymer solution viscosity after EO incorporation, as well as good antibacterial properties against *Staphylococcus aureus* for all tested oils. The obtained data underlined the fibers potential to be used for wound dressings and in tandem for synergism with antibiotics.

Another crosslinked oil-sphere complex system, based on calcium-aluminum-alginate-pectinate encapsulated with *Mentha piperita*, was investigated for the treatment of irritable bowel syndrome [82]. The study put into evidence the dependence of the physicochemical and textural properties of the prepared complex on the concentration between polymers, crosslinkers, and crosslinking reaction times. Moreover, the resulted kinetic modeling data indicated the diffusion process as the predominant release mechanism of the encapsulated oil, at the same time as modulating the diffusion through the application of a novel fusion coating procedure. The researchers used the artificial neural network as an alternative tool to predict the response values of the dependent variables, and concluded that this approach is very attractive when employing statistical strategies for formulation development. Thus, the Plackett–Burman design was employed to develop and optimize the new crosslinked calcium–aluminum–alginate–pectinate oil sphere complex for the in vitro site–specific release of *Mentha piperita*. The encapsulation of oil ranged from 6 to 35 mg/100 mg oil spheres.

*Mentha* (the Lamiaceae family), which was extensively studied for its biological actions, was reviewed in terms of the antioxidant, antifungal, antibiofilm, and cytotoxic properties of the *Mentha spicata* (*Mentha spp.*) oil [83]. The article provided comprehensive information about its use in the treatment of fungal infections, or as antioxidants and integrative anticancer therapy, and about its effectiveness in treating diseases without causing any serious adverse reactions. In another study, which also supports the utilization of *Mentha spp.* EO as a traditional medicine, there are opened the perceptions for more potent substances in mentha oil for the management of obesity, Alzheimer’s disease, and dermatophytosis, and for combating drug-resistant bacterial infections [84].

#### 2.1.2. Cellulose Derivatives

Cellulose has attracted attention due to its wide availability, low-cost, biocompatibility, and biodegradability, and due to its capacity for derivatives preparation. It has been widely used as a matrix for EO encapsulation.

Among these derivatives, cellulose acetate presents biocompatibility, absence of toxicity, has biodegradability, and has recently been used for the incorporation of lemongrass oil via the solvent/anti-solvent method, to prepare antimicrobial nanocapsules (NCs) [85]. The new system prepared and investigated by Liakos et al., demonstrated exceptional antimicrobial properties due to lemongrass oil, characteristics that recommend the compound as a pharmaceutical agent for biomedical applications. The resulting nanocapsules diameter was tailored between 95 and 185 nm, which bio-adhered well to mucous membranes, and presented very good antimicrobial properties at low concentrations against *Escherichia coli* and *Staphylococcus aureus*. An important aspect of this new structure is the absence of surfactants during nanocapsules preparation, which makes the resulting NCs fully natural. The formation of the chemical bonding between lemongrass and the cellulose acetate ring with hemiacetal structure formation, which ensured the stability of the lemongrass oil molecules in the aquatic environment for more than 6 months, was also demonstrated.

Cellulose acetate was investigated as a carrier for the controlled release of thymol, a natural substance with strong antibacterial properties [86]. The impregnation of thymol was realized by using high-pressure techniques, and the thymol presence was confirmed by various techniques. For the determination of the release kinetic of thymol, simulated gastric and intestinal fluids were used, and the results were correlated with Korsmeyer–Peppas and Weibull models. The new products showed antibacterial activity against 23 tested bacterial strains, including methicillin-resistant *Staphylococcus aureus*.

Another cellulose derivative, in which some hydroxyl groups present on the glucose moiety were modified into ethyl ether groups, namely, ethyl cellulose was used for Babchi EO encapsulation due to the non-swellable and hydrophobic character of the EO [87]. Babchi EO possess a variety of biological activities, such as antitumor, anti-inflammatory, immune-modulatory, antioxidant, antifungal, antimicrobial, and antibacterial properties, which recommend its use for the treatment of dermatological disorders. Due to its highly viscous nature and poor stability in the presence of light, air, and high temperature, its practical applications are very limited. In the mentioned study, Babchi EO was encapsulated in the ethyl cellulose matrix through a quasi-emulsion solvent evaporation technique by using PVA as stabilizer and dichloromethane as solvent. The in vitro cytotoxicity revealed the prepared microsponges as safer products on dermal cells, with antimicrobial activity against dermal bacteria like *Staphylococcus aureus*, *Pseudomonas aeruginosa*, and *Escherichia coli*, data that confirmed enhanced antibacterial behavior.

As already mentioned, the electrospinning nanofabrication is a very useful technique for EOs encapsulation in a suitable polymer carrier as it does not require the use of coagulation chemistry or high temperatures to produce solid threads from solution. As a result, many research studies were conducted for the encapsulation of EOs in polymeric electrospun fibers to create functional membranes for biomedical and food packaging applications. Additionally, these studies underlined the preservation of the bioactivity of EOs after mixing with various polymer systems and processing by electrospinning. Moreover, the EOs released from the products obtained through the electrospinning process still presented the characteristics of the inserted EOs, namely antimicrobial, anti-inflammatory, and antioxidant activity [88].

Liakos et al. reported the use of EOs as natural antimicrobial agents for the preparation of cellulose-based fibrous dressings. They produced composite electrospun fibers for encapsulation of cinnamon, lemongrass, and peppermint oils, and demonstrated the capacity of the fibers to inhibit the growth of *Escherichia coli.* It was also proved from assays on skin cell models that the fibers are biocompatible and not cytotoxic [89]. The fabrication of the fibrous mats by electrospinning necessitates specific conditions to obtain mats with thickness of about 0.2 mm. The presence of the EOs in fibers was proved by Raman spectroscopy, while the antibacterial and the efficacy of the fibers was tested and confirmed against *Escherichia coli*, resulting in a complete inhibition of bacteria growth. The biocompatibility assays performed on two different cell lines (fibroblasts and human keratinocytes) revealed the non-cytotoxicity of the scaffolds.

The same polymeric system and electrospinning technique were used to valorize and test the antimicrobial properties of rosemary and oregano oils [90]. The prepared electrospun fibers, based on cellulose acetate and various content of EOs, proved to have good antimicrobial properties against the bacteria species *Staphylococcus aureus*, *Escherichia coli*, and the yeast *Candida albicans*. Among the selected EOs, the electrospun fibers prepared with oregano oil showed the best antimicrobial and anti-biofilm effects due to the potency of this oil against bacteria and fungi, especially for *Escherichia coli* and *Candida albicans*.

Rosemary’s beneficial properties are well known, such as analgesic, antinociceptive, antidepressant, anxiolytic, anti-inflammatory, antidiabetic, anti-neurodegenerative, and use to alleviate rheumatic pain, stomachache, and dysmenorrhea [91]. Moreover, recently, the synergistic antinociceptive interaction of the ethanolic extract of rosemary officinalis and the *Syzygium aromaticum* EO, suspended in 0.5% Tween 80–0.9% isotonic saline solutions, co-administered with ketorolac solution in rats, was demonstrated. Data were supported by a combination of conventional and alternative therapies [92]. In other studies, it was indicated that extracts and purified components of carnosic acid and rosemary displayed significant growth inhibitory activity on a variety of cancers [93]. The investigation confirmed the ability of rosemary/carnosic acid to inhibit the growth of human breast cancer cells and to synergize with curcumin, which inhibits the proliferation of ER-negative human breast cancer cells and induces G1 cell cycle arrest.

Reports concerning the antitumor activities of *Rosemary* (*Rosmarinus officinalis L*.) registered both in vitro and in animal studies, as reviewed in the article by Gonzalez-Vallinas et al. [94]. If initially, the antitumor effects of rosemary were attributed to its antioxidant activity, the contemporary studies present different molecular mechanisms related to rosemary tumor inhibitory properties, for example, the diterpenes from rosemary interact with the cell signaling machinery of cancer cells leading to decreased cell viability and proliferation [95]. Thus, the review summarizes the anticancer effects of rosemary developed through molecular mechanisms, as well as the interactions between rosemary and currently used anticancer agents, underlining the possibility to use rosemary extract as a complementary agent in cancer therapy in comparison with its isolated components. The pharmacokinetic properties of diterpenes from rosemary suggest they are well absorbed and can achieve plasma concentrations that are physiologically relevant, while the phenolic components of rosemary possess anti-invasive and anti-metastatic activity.

A review providing an overview of the anti-cancer compounds derived from plants showing promising anti-carcinogenic effects against various skin cancer cell lines and on animal models was recently published [96]. Phytochemicals, as for example steroids, coumarins, terpenes, EOs, alkaloids, esters, ethers, resins, phenols, and flavonoids developed in marketed formulations for skin cancer prevention and treatment, were presented.

Among the tested EOs, the emulsified *Brucea javanica* oil, which has been used in treating different kinds of malignant tumors, was investigated, as well as the *Brucea javanica* oil emulsion variant, enriched with brusatol, using soybean lecithin surfactant, for testing their effects against hepatoma 22 (H22) in mice [97]. The investigation was conducted based on previous studies concerning *Brucea javanica* that demonstrated the antitumoral, antimalarial, and anti-inflammatory properties of EO, including its use in the treatment of lung and gastrointestinal cancers. The antitumor components were tetracyclic triterpene quassinoids, with poor water solubility and low bioavailability, which necessitated increased investigations, to increase the clinical applications and proper networks for encapsulation [98]. The study by Tongtong Wang et al. [97] confirmed brusatol, a characteristic quassinoid, as the major category of anticancer phytochemicals of *Brucea javanica*. The compound exhibited pronounced anti-hepatocellular carcinoma (anti-HCC) activity confirmed by suppressing the growth of implanted hepatoma H22 in mice, ascending weight, abdominal circumference, ascites volume, and cancer cell viability, while the anti-HCC effect was associated with the activation of miRNA-29b, p53-associated apoptosis and mitochondrial-related pathways.

#### 2.1.3. Chitosan

Chitosan (CS), which is a natural polymer, was frequently used in tissue engineering and wound healing due to its intrinsic antimicrobial, hemostatic, non-toxic, healing-stimulant, and biodegradable and biocompatible properties [99]. It was also used in matrix formulations for EO encapsulation.

Thus, in a research study, thymol oil was embedded in CS, a mucoadhesive polymer, processed in the form of hydrogels, and the investigations confirmed the advantages of the new compound for a biomedical application [100]. The in vitro results indicated biocompatibility when exposed to [3T3] fibroblasts and also that they have antimicrobial activity against *Staphylococcus aureus* and *Streptococcus mutans* for 72 h, and antioxidant activity for 24 h. The resulting data constitute desirable properties for a mucosal delivery system in antimicrobial–antioxidant dual therapy for periodontal disease.

The antioxidant and free radical scavenging activity and the antihypertensive effect of the aqueous *Thymus serpyllum L*. extract was proved by in vitro investigations [101]. Thyme polyphenols, obtained by diffusion from an external aqueous solution of *Thymus serpyllum L*., were encapsulated into CS microbeads crosslinked by the emulsion technique [102]. The conditions of loading by swelling of dry microbeads in the accommodated acidic medium were correlated with the concentrations of chitosan (1.5–3% (*w/v*)) and the crosslinking agent, namely glutaraldehyde (0.1–0.4% (*v/v*). The encapsulated thyme polyphenols had a rate of release in simulated gastrointestinal fluids prolonged with 3 h, and the new system has the potential to be used as a functional food additive.

Thyme and carvacrol EOs were embedded in CS nanoparticles by nanoprecipitation procedure from acetic acid, or in nanocapsules prepared through a solvent displacement technique [103]. The kinetics of the release of thyme EO in aqueous medium from the prepared CS nanostructures were evaluated. The inhibitory activity assess for future applications as an antibacterial agent of the compound was put into evidence, and shows a higher inhibitory activity for nanoparticles than for nanocapsules when tested against six foodborne bacteria (the highest against *Staphylococcus aureus* in case of nanoparticles, and highest against *Bacillus cereus* in case of nanocapsules). At the same time, thymol and carvacrol, presented faster release time from nanoparticles than from nanocapsules.

In a recent investigation, it was hypothesized that the association of rosmarinic acid-loaded nanoemulsions with CS would enhance the EO stability, its mucoadhesive properties, the penetration and/or retention, and residence time in the nasal mucosa [104]. The new formulation was intended to be used as a new approach in neuroprotective therapies and in this context, the cell viability, and death in the MRC-5 fibroblast cell line were evaluated. An optimization program for the preparation of the mucoadhesive chitosan-coated nanoemulsions was developed, and based on this program, the optimized conditions were found, namely 8.5% oil phase (*w*/*v*), 3:10 lecithin to oil phase ratio (*w*/*w*), and 0.1% final CS concentration (*w*/*v*).

The synthesis of β-cyclodextrin (CD) modified CS (β-CD/CS) nanoparticles as a binary system via the ionic gelation method, was used for the encapsulation and the controlled release of *Cinnamomum zeylanicum* EO [105]. In this research study, the investigated EO, which is an effective antimicrobial agent, followed a Fickian behavior regarding the release mechanism from β-CD/CS nanoparticles, while the analysis of its in vitro release showed a sustained and controlled release for over 120 h. It was concluded that the as-synthesized β-cyclodextrin/chitosan nanoparticles are promising systems for enhancing the therapeutic effect of the *Cinnamomum zeylanicum* EO.

CS as a natural and biodegradable cationic polysaccharide with efficient antimicrobial action, together with PVA, were used to form a film embedded with cinnamon leaf and clove oils that display, as well, strong antimicrobial activity, especially against *Staphylococcus aureus* and *Pseudomonas aeruginosa* [106]. The prepared CS/PVA films embedded with cinnamon leaf or clove oils demonstrated to be mechanically fit according to the investigation, presented efficient bactericidal effects after 2 h of direct contact in the infected microenvironments, and evidenced improved antimicrobial character compared with the unloaded films.

An emulsion-ionic gelation procedure of CS crosslinked with pentasodium tripolyphosphate (TPP) or sodium hexametaphosphate (HMP), as crosslinkers, was investigated. The resulted CS nanoparticles were loaded with *Carum copticum* EO [107]. From the investigation, it was concluded that the encapsulation efficiency and loading capacity of *Carum copticum* EO in CS nanoparticles increased with the increase of the oil amount. Moreover, concerning the obtained nanoparticles, they were spherical in shape and with a regular distribution, while the in vitro release profiles exhibited an initial burst release of the oil followed by a sustained release at different pH conditions, respectively.

CS was also used in a study for the preparation of nanoparticles able of encapsulation of peppermint oil and green tea oil, both known for their nutritional and biomedical properties [108]. The method used for EO encapsulation was also emulsification/ionic gelation, and the scheme representing the preparation procedure, shown in Figure 3, is illustrative for EO nano-encapsulation. Reproduced with permission [108].

The resulted nanoparticles were characterized and a loading capacity of about 22.2% for peppermint oil, and respectively 23.1% for green tea oil, was revealed. The new nano-encapsulated particles presented an increased thermal stability, as well as antioxidant activity by ~2 and 2.4-fold for peppermint oil and green tea oil respectively. In addition, the antibacterial activity of CS/green tea oil NPs was more efficient than CS/peppermint oil NPs especially against *Staphylococcus aureus* with ~9.4 folds improvement compared to pure green tea oil, and ~4.7 fold against *Escherichia coli*.

Underlining the EOs alternative to the current antibiotics due to their potent and broad-spectrum antimicrobial potential, unique mechanisms of action and low tendency to induce resistance, Bushra Jamil’group explored the anti-pathogenic potential of some EOs in a bio-based nanocarrier system [109]. Cardamom oil was selected for nanoencapsulation because of its most potent antimicrobial activity, and it was loaded in CS nanoparticles by ionic gelation method with an encapsulation efficiency of more than 90% in nanoparticles evidencing dimensions of about 50 to 100 nm. The investigation indicated the preparation of stable nano-dispersion with Zeta potential higher than +50 mV, while the cytotoxicity analysis of the obtained nanoparticles proved non-hemolytic and non-cytotoxic behavior on human corneal epithelial cells and HepG2 cell lines. The new nano-encapsulated systems exhibited excellent antimicrobial potential against extended spectrum of β—lactamase producing *Escherichia coli* and methicillin-resistant *Staphylococcus aureus*, data which recommend the new nanoparticles for treating multidrug-resistant pathogens, and hence offer an effective alternative to current antibiotic therapy.

The new method used for encapsulation based on emulsion–ionic gelation crosslinking, constitutes a promising technique for the entrapment of bioactive compounds, which does not require heat application or the use of toxic crosslinking agents and generates stable nanosized particles. This two-step method was used in a research study for the fabrication of carvacrol-loaded CS nanoparticles [110]. Thus, the oil-in water emulsion of CS followed by ionic gelation in the presence of TPP led to preparation of particles able for encapsulation of carvacrol in a content of 0.25 up to 1.25 g/g of CS. The carvacrol-loaded CS nanoparticles were tested from the viewpoint of their antimicrobial activity against *Staphylococcus aureus*, *Bacillus cereus* and *Escherichia coli*, and presented a MIC of 0.257 mg/mL. The release mechanism of carvacrol from CS nanoparticles followed a Fickian behavior with a plateau level on day 30, and specific release amounts in interdependence with the pH of the used solutions.

It must be mentioned that carvacrol possesses various pharmacological properties, especially in the treatment of cardiovascular diseases and state-of-the-art studies on its antimicrobial, antioxidant, and anticancer properties were recently reviewed [111,112]. In this context were mentioned (`) the anticancer properties in preclinical models of breast, liver, and lung carcinomas, where is acting on proapoptotic processes, (``) the effectiveness against food-borne pathogens as for example *Escherichia coli*, *Salmonella*, and *Bacillus cereus*, (```) the high antioxidant activity especially when it is associated with thymol. Moreover, the investigations proved a reduction in the progression of arterial hypertension in spontaneously hypertensive rats, and the pharmacological potency to phenylephrine in tested groups.

CS–TPP nanoparticles were encapsulated with eugenol (4-allyl-2-methoxyphenol) a naturally occurring phenolic compound with applicability in pharmaceutical, cosmetics, food, and active packaging domains due to its effective antimicrobial and antioxidant properties [113,114,115]. Eugenol-loaded CS nanoparticles were prepared following the same two-step procedure based on emulsion–ionic gelation crosslinking. The researchers first prepared the oil-in-water (o/w) CS emulsion in the presence of Tween 60 as emulsifier, over which in the second step a variable eugenol and TPP content were dropped. The obtained particles were collected by centrifugation and characterized from the viewpoint of eugenol encapsulation capacity and their improved thermal stability. The new matrix system and method used for eugenol encapsulation allowed a loading capacity (LC) and an encapsulation efficiency (EE) in the ranges of 0.40–12.8% and 1.1–20.2%, respectively, for an initial content of eugenol of 50–125% (*w/w*), while CS and TPP concentrations were of 0.5% and 1.0% (*w/v*), respectively. An increased content of TPP did not improve the LC or EE for the prepared particles. In addition, improved thermal stability of encapsulated eugenol was registered compared with nude eugenol, as well as greater radical scavenging activity.

The two-step procedure of oil-in-water emulsion preparation followed by the ionic gelation of CS with TPP was also used for oregano EO encapsulation in the polysaccharide nanoparticles [116]. The EO encapsulation was confirmed by specific spectrophotometric and thermogravimetric analysis. The EE and LC of oregano EO loaded in CS nanoparticles were found around 21–47%, and 3–8%, respectively, corresponding at an initial EO content of 0.1–0.8 g/g CS. The performed in vitro release studies indicated an initial burst effect followed by a slow release of EO.

Electrospun nanofibrous membranes, which present high surface area to volume ratio, porosity, and structural similarities with the skin extracellular matrix, meet the requirements for wound dressing’s applications. As shown in the article written by Miguel et al., an ideal wound dressing must confer protection against external microorganisms, chemical, and physical aggressions, and to promote healing process by stimulating the cell adhesion, differentiation, and proliferation, and the nanofibers incorporated with EOs bioactive molecules, correspond to these requests [117].

Solutions based on CS and poly(ethylene oxide) (PEO) were incorporated with cinnamon-aldehyde, a volatile EO that eradicates pathogens non-specifically [118]. The obtained solutions were electrospun into mats with ~50 nm fiber diameters. Cytotoxicity studies have proven intrinsic antibacterial activity and quick release of cinnamon–aldehyde, and enabled high inactivation rates against *Escherichia coli* and *Pseudomonas aeruginosa*. It was concluded that the prepared nanofiber mats could be used as flexible scaffolds to alleviate nosocomial infections by delivering a broad-spectrum of natural antimicrobial agents.

The electrospinning procedure was used as well to produce continuous nanofiber mats for wound dressings from solutions based on a combination between natural and synthetic polymers, namely CS, PVA, and gelatin (CS/PVA/Gel), which were incorporated with *Zataria multiflora* EO, a strong natural antimicrobial agent [119].

The new realized nanofiber mats, loaded with 10% EO, and chemically crosslinked by glutaraldehyde vapors, presented antimicrobial activity determined by the Amphibious Air Traffic Control Center (AATCC100) method. Thus, it was proved the completely inhibition for the growth of *Staphylococcus aureus*, *Pseudomonas aeruginosa*, and *Candida albicans* after 24 h of incubation. The nanofiber mats also exhibited suitable swelling and mechanical properties, and, according to the investigation results, may be recommended as wound dressings in surgery and burn wounds.

The use of natural polymers for fabrication of biomedical electrospun scaffolds incorporated with bioactive molecules as therapeutic agents to generate innovative bioactive wound dressings with improved potential of healing, was also recently reviewed [120].

#### 2.1.4. Starch and Maltodextrin Based Systems

A freeze-drying encapsulation technique was used for microencapsulation of rosemary EO, known through its monoterpenes constituents like 1,8-cineole, by means of two coating materials, namely maltodextrin (MD) and whey protein concentrate (WP) in different coating ratios [121]. MD was chosen due to its excellent oxygen-blocking property, while WP, the second coating material, shows low viscosity even in high concentrations, and represents an excellent emulsifier. The optimum coating material formulation was obtained after investigating the physicochemical properties and storage stability of microcapsules. The emulsions were analyzed for their particle size distributions, and the freeze-dried capsules for their encapsulation efficiency, surface morphology, and concentrations of 1,8-cineole during storage. The performed encapsulation allows the use of rosemary oil for medical antiseptic and astringent purposes.

Chemically modified MD has been investigated for the microencapsulation of avocado EO, owing to its capacity to protect bioactive compounds [122]. MD modified with octenylsuccinic anhydride presented a degree of substitution of 0.020, and an improved emulsification capacity compared to its non-modified counterpart. The microencapsulation capacity of the modified MD was tested with avocado EO by the spray drying technique. The modified MD confirmed its potential for EO protection through an increased stability put into evidence by the reduction of about 60% of the peroxide index relative to MD alone.

A gum acacia/MD polymeric blend was investigated for the microencapsulation of lavender oil by using a spray drying technique in order to protect the 100 oil components, which are susceptible to volatilization and degradation reactions [123].

The EOs microencapsulation obtained by the spray drying technique was also reviewed in relation to the principal EOs reported in the literature, their compositions, and properties, as well as in terms of antibacterial and antioxidant character, but also the parameters used in the microencapsulation processes [124].

The anti-inflammatory ability and wound-healing properties of lavender oil are well known, and Burhan et al. used this oil in their study and made a correlation between polymer content, oil loading, and the ratio between gum acacia and MD polymeric matrix. The conditions for microparticles preparation were reflected in their particle size, yield, loading, and encapsulation efficiency of composites. It was found that the lavender microparticles with a size of 12.42 ± 1.79 mm were prepared at 30 *w/w*% polymer concentration, giving a 16.67 *w/w*% oil loading, while the 25 *w/w*% gum acacia showed maximum oil protection at a loading capacity of 12 mg *w/w*%, and an encapsulation efficiency of about 77.89 *w/w*%.

Arabic gum, MD, and modified starch system was used to encapsulate oregano EO by spray drying after preparation of the oil-in-water emulsions [125]. The polysaccharides functioned as emulsion and EO stabilizers, and also as tablet binders. The spray drying procedure gave a powdered product with various content of oil. The transformation in tablet of the powdered product was made by addition of (1) croscarmellose sodium that contributed at the increasing of the release rate of EO from tablets, or (2) sodium starch glycolate having small effects on the release process. The tablets with oregano oil showed antimicrobial activity comparable with the non-encapsulated EO form. The authors of study recommended the tablet formulation as a promising dosage form for oral delivery of EO providing an accurate dose that can be delivered in a convenient way.

Starch as biopolymer is often used in pharmaceutical formulations due to its biodegradability and versatility [126,127]. Modified starch had been also investigated for encapsulation of EOs in various forms, for example, spherical aggregates, nanoparticles, or porous compounds [128,129,130,131]. In a recent investigation, the nanoencapsulation of clove essential oil in 3D nano-network porous starch-based material to enhance the antimicrobial activities of the matrix, it was pursued [132]. The study presented an elaborate preparation process that included starch gelatinization with subsequent complexation (and further remove) of the CaCO_3_ template with 3D-nanoparticles formation for the clove EO encapsulation. The new nanoparticles containing EO presented enhanced and prolonged bacteriostatic effect. The schematic illustrative preparation of the nanoparticles was presented by the authors (Figure 4).

A recent study investigated the preparation of composite films based on sago starch, guar gum, and whey protein isolate with antimicrobial properties due to incorporation of carvacrol, and citral, as well as EOs combination, in order to be used for the prophylaxis of the bacterial gastroenteritis [133]. The oils droplets were uniform distributed within the biopolymeric network, and all films containing EOs demonstrated good antibacterial potency against the bacterial gastroenteritis model causing bacteria, namely, *Bacillus cereus* and *Escherichia coli*. As a result, the researcher concluded about the possibility of using the prepared antimicrobial films for the prophylaxis of bacterial gastroenteritis.

The encapsulation of thymol in the potato-starch–polysorbate–glycerol–citric acid complex dispersion and related cast films was demonstrated [134]. The results proved the antioxidant and antimicrobial capacity of potato starch dispersions enriched with polysorbate-thymol micelles, only in the presence of polysorbate-thymol micelles, but with lower levels than polysorbate–thymol alone. It was concluded on the necessity to evaluate the biocompatibility of these materials.

#### 2.1.5. Whey Protein

Another direction where the antimicrobian character of EOs has found applicability is against fungal metabolites that frequently co-occur in foodstuffs and are responsible for mycotoxicosis and several primary cancers. In this context, a study was devoted to the use of cinnamon EO as emulsion droplets encapsulated in whey protein as wall material, for the protection against oxidative stress, cytotoxicity, and reproductive toxicity in male Sprague-Dawley rats subchronically exposed to Fumonisin B1 (FB1) and/or aflatoxin B1 [135]. In the wall material, which was constituted from whey protein concentrate (10% in water), it was encapsulated cinnamon EO in a ratio of 1:4 (wall/core material) by stirring and homogenization by ultrasonication, and further the prepared emulsion was spray dried. The analyses performed onto the prepared dispersion evidenced an average size of 235 ± 1.4 nm and zeta potential of −6.24 ± 0.56, while the GC–MS analysis revealed the presence of cinnamaldehyde, α-copaene, trans-cinnamaldehyde, caryophyllene, and delta-cadinene, which are the main compounds of cinnamon essential oil. The investigation proved that the prepared dispersion did not induce toxic effects. Moreover, it could normalize the majority of the tested parameters and improve the histological data obtained from in vivo tests. These effects were attributed to the antioxidant effect of the dispersion containing cinnamon EO in synergy with the whey protein concentrate antioxidant effect, as well as to cinnamon EO dispersion ability to enhance the gut microflora growth, determine the adsorption of both mycotoxins and decrease their bioavailability.

#### 2.1.6. Silk Fibroin

In another investigation, the electrospinning technique was used to obtain an electrospun asymmetric membrane composed of two interconnected layers designated for enhancing healing processes [136]. The top layer was composed of silk fibroin and polycaprolactone (PCL), while the bottom layer was produced by using a blend of silk fibroin with hyaluronic acid. Silk fibroin was chosen as it presents excellent biocompatibility, good water vapor permeability, biodegradability, mechanical strength, and minimal inflammatory reaction [137], while PCL was selected owing to its hydrophobic character and high mechanical strength [138]. It was intended that both compounds to generate an epidermis-like layer presenting hydrophobic character, waterproof ability, and mechanical resistance. The researchers have also chosen hydrophilic hyaluronic acid as it provides high capacity of hydration, water-sorption and water retention, allowing cell attachment, migration, and proliferation [139,140]. The resulting data highlighted the epidermis-like structure of the new generated “sandwich”, which was capable of absorbing the wound exudate and also promote cell adhesion and proliferation. The antimicrobial activity of the bottom layer was attributed to the thymol EO loaded into the nanofibers mesh.

#### 2.1.7. Gelatin

*Piper aduncum* L. and *Piper hispidinervum* C. EOs were encapsulated in gelatin nanoparticles, in the presence Tween 80, PCL, Span 60 and caprylic/capric triglyceride acid during the homogenization process [141]. The prepared nanoparticles were tested against *Aedes aegypti* Linn., *Tetranychus urticae* Koch., and *Cerataphis lataniae* Boisd. The investigation proved an encapsulation efficiency of the oils up to 1000 μg × mL^−1^, while their controlled release was described by the anomalous mechanism of Korsmeyer–Peppas.

A biocompatible patch, based on gelatin film reinforced with bacterial cellulose polyelectrolyte, was loaded with curcumin and tested from the viewpoint of the wound healing activity [142]. The obtained patch confirmed antimicrobial activity assessed via Field Emission Scanning Electron Microscopy (FESEM) analysis and live-dead assay using propidium iodide and 4′,6-diamidino-2-phenylindole as fluorescent indicator. Based on the resulting data, the researchers recommended the new prepared hydrogel patch as a new direction in the transdermal drug delivery systems.

### 2.2. Synthetic Macromolecular Structures for EOs Encapsulation

In addition to natural polymers, the biocompatible and biodegradable synthetic ones constitute an effective alternative to scaffold materials that couple, embed, and/or encapsulate the EOs to be applied in medicine and pharmaceutical sciences.

Various biodegradable and biocompatible synthetic polymeric compounds have been electrospun into nanoscale fibers to be further used in the biomedical field including as carriers for drug delivery. From this group PCL, polylactic acid (PLA), polyurethane, and polyvinyl pyrrolidone (PVP) are the most currently applied owing to their biodegradable nature, hydrophilicity, chemical, and thermal stability. They have as well good tissue compatibility, solute permeability, and excellent electrospinnability, and, as a result, they were intensively investigated as matrix for various bioactive substances including EOs [143,144].

For example, PCL and PLA, as well their 50/50 polymeric systems were used for mats preparation by electrospinning technique [145]. Thymol, in concentration of 1.2% *v/v*, was embedded in the obtained nanofibers, and the prepared compounds were tested for wound dressings. In the process of electrospinning, the researchers used PCL (12% *w/v*) and PLA (3% *w/v*) in chloroform/dimethylformamide solution in ratio of 7:3. The antibacterial evaluation of the 50/50 PCL/PLA hybrid nanofibrous samples containing thymol showed satisfactory effects on *Staphylococcus aureus* compared to *Escherichia coli* bacteria during treatment periods. In vivo wound-healing in rats and the histological performance observations of the nanofibrous mats made of 50/50 PCL/PLA loaded with thymol revealed a wound-closure percentage of about 92.5% after a period of 14 days.

Composite nanofiber dressings realized from polyurethane encasing lavender oil and silver (Ag) nanoparticles were tested as multifunctional wound dressings [146]. The investigations made on the prepared nanofibers have been shown to provide protection against external agents as well as promote the regeneration of new tissue. The presence of Ag NPs and lavender oil improved the hydrophilicity of the nanofibers and ensured the proliferation of chicken embryo fibroblasts cultured in vitro, as well as the antibacterial efficiency against *Escherichia coli* and *Staphylococcus aureus*. The authors presented a suggestive scheme illustrating the nanofiber composites fabrication through electrospinning, as well as their multifunctional character for cell growth and antimicrobial behavior (Figure 5).

Nanofibrous membranes, as highly soft materials with high surface-to-volume ratios based on PCL/PVP system, were prepared to be used as carriers for therapeutic agents with antibacterial properties to accelerate wound healing [147]. Crude bark extract of *Tecomella undulata*, a medicinal plant known for its ability in medical applications for wound healing, was added to a polymer solution in chloroform/methanol (4:1) mixture, made of PCL (which offers strength to the matrices) and PVP (which constitutes the drug delivery medium). The new system was subjected to the process of electrospinning and the resulted nanofibrous membranes were tested against standard strains of *Pseudomonas aeruginosa* MTCC 2297, *Staphylococcus aureus* ATCC 933, *Escherichia coli* (IP-406006). The investigation confirmed the ability of the new PCL/PVP nanofiber mats to inhibit the growth of the bacterial strains, indicating also the potential of the system for use as a wound dressing.

PCL was used as well in tandem with PVA in a coaxial electrospun procedure for the fabrication of core–shell nanofibers incorporated with thyme extract [148]. The effect of operational parameters for the preparation of the nanofibers was presented in order to obtain nanofibers with uniform surface morphology and acceptable tensile strength. It was also investigated and confirmed the biocompatible character of the nanofibers after thyme encapsulation, whereas the antibacterial activity was validated against two bacteria—gram-positive, namely *Staphylococcus aureus*, and gram-negative *Escherichia coli*, for an encapsulation of 5% (*w/v*) thyme extract. The potential of the new core–shell PCL/PVA nanofibers for wound healing and for the treatment of surfaces that contain pathogenic microorganisms was proved as well.

PCL and PCL-hydroxyapatite composites were used for the preparation of functional porous scaffolds after impregnation with thymol by using a supercritical solvent procedure [149]. The new complex was investigated as an environmentally friendly product, while the prepared functional scaffolds, with controlled microstructure dependent on the conditions of impregnation, were put into evidence.

Recent studies presented the incorporation of EOs into electrospun fibers not only to confer bioactivity but also to control some physical characteristics of the resulting mats [150]. The use of linalool in therapeutics, cosmetics, and antimicrobial and larvicidal products is well known and as result its encapsulation for further controlled release was investigated. In a study of Souza team three different concentrations, namely 10, 15, and 20 wt. %, of linalool oil were incorporated in PLA nanofibrous membranes by electrospinning technique and by solution blow spinning procedure [151]. The average diameters of the new nanofibers were similar, ranging from 176 to 240 nm in both cases of preparation. The plasticizer effect of linalool was put into evidence by the decreased values of the glass transition temperature, melting point and crystallization temperature of the resulted composite. The oil release from PLA nanofibrous membranes followed a Fickian rule. The solution blow spinning procedure proved a better encapsulation of linalool concretized in increased time for release. Increased hydrophobicity of the PLA nanofiber membranes was obtained by both preparation procedures.

In a study by Mori et al., various concentrations of Candeia (*Eremanthus erythropappus*) oil were incorporated into PLA, and the nanofibrous mats were produced by the electrospinning procedure [152]. The oil addition increased the nanofiber diameter, while the homogeneous structure of the new prepared nanostructured mats was evidenced by Scanning Electron Microscopy (SEM). The plasticizer effect of Candeia was confirmed by the decreased values of the glass transition temperature and melting point corresponding to the new PLA composite, and was attributed to the presence of alpha-bisabolol as main terpene compound in oil. The prepared nanofibers demonstrated antibacterial and hydrophobic characters, and were recommended by authors for applications in controlled release of drugs.

Tea Tree and Manuka EOs are known for their biological activity and utilization in aromatherapy or for the treatment of microbial infections, and due to their special properties were tested for encapsulation by electrospinning in PLA fibers [153]. The effects of various concentrations of EOs (from 2.5 up to 15.0 % *v*/*v*) on the fiber’s morphology, and on the thermal and mechanical properties of fibers, was investigated. The data confirmed the plasticizer potential of both EOs, resulting in lower glass transition temperature up to 40 °C as compared to the one of pristine PLA fibers. Both elongation at break and tensile strength at break increased after increasing the amount of EOs with values up to 12 times higher than those of PLA fibers without EOs. The antibacterial activity corresponding to PLA/Manuka fibers was concretized in inhibition of the *Staphylococcus epidermidis* growth, whereas PLA/Tea Tree EO mats were ineffective in preventing the biofilm formation.

A review concerning PLA-based antimicrobial materials was recently published [154]. The article underlined the physicochemical properties of PLA as renewability, biodegradability, U.S. Food and Drug Administration (FDA) approval for clinical use, which recommends this compound as promising system to control microbial growth. The possibilities for the development of antimicrobial PLA materials as delivery systems, coatings for biomedical devices, efficient stabilizers-carriers of various kinds of antimicrobial additives including EOs, and other natural compounds, active particles, nanoparticles, and conventional and synthetic molecules, were also mentioned.

In another review, by Roberto Scaffaro, some methods used for carvacrol incorporation into the PLA matrix, in relation to the morpho-mechanical properties, release behavior, and antimicrobial activity of the new composite systems, were critically presented [155]. The article presented the potential of PLA/carvacrol composite for the development of drug delivery systems, in the coating of medical devices, or in food packaging. Various methods for carvacrol incorporation into PLA and processing techniques, as for example by melting, solution, or hybrid, which included two- or multi-step protocols, involving both solvent-based and melting-based steps, were also presented.

PLA, poly(methyl methacrylate) (PMMA) and PLA/PMMA blends were investigated for use as matrix loaded with EOs via the solvent evaporation technique [156]. The obtained polymeric microspheres, with average size between 1.5 to 9.5 mm, were loaded with linalool, 4-allylanisole, and trans-anethole. The highest encapsulation was registered in systems consisting of linalool, namely 81.10 ± 10.0 wt. % for PLA system, and 76.0 ± 3.3 wt. % for PMMA system, while the release rate of EOs from the microspheres was affected by the system size. The composite microspheres were tested for antibacterial activity against both Gram-negative and Gram-positive bacterial strains, and exhibited inhibitory effects against *Escherichia coli* and *Staphylococcus aureus*, and proved capacity as antibacterial agents.

Electrospinning as feasible nanotechnology was used for poly(acrylonitrile) (PAN) nano-fibrous mats preparation with entrapped lavender oil [157]. During the electrospinning process the incorporation of an electrolytic solution of 0.3% (*w/w*) NaCl to the polymer solution led to obtaining a reduced average fiber diameter (88.44 nm), with a narrow degree of polydispersity, and enhancement of the fiber morphology, which improved the probability of incorporating therapeutic oils into nanofibers. The prepared PAN composite nanofibers were recommended for use in various fields, such as biomedical, textile and water treatment applications, owing to their proven antibacterial characteristics against *Staphylococcus aureus* and *Klebsiella pneumoniae* bacteria. PAN nanofibers presented good biocompatibility and non-cytotoxic nature tested on mouse fibroblasts.

Recently, the obtainment of antibacterial nanofibers by an oil-in-water emulsion electrospinning procedure was presented. The system was based on PVP and cinnamon oil [158]. The characterization of the new system evidenced its capacity for electro-spinning process, while the antimicrobial properties of the prepared nanofibers were confirmed by the disc diffusion method against *Staphylococcus aureus*, *Escherichia coli*, *Pseudomonas aeruginosa*, and *Candida albicans*. It was found a correlation between the increase of the viscosity values of the polymer/oil complex after addition of a surfactant, and in interdependence with the increase of the cinnamon concentration, as well as of the average fiber diameter or the spinnability and fiber smoothness.

Self-nanoemulsifying drug delivery system (SNEDDS) formulations in water, based on *Piper cubeba* EO and synthetic and nontoxic components, namely Sefsol-218 (a propylene glycol caprylate), Triton-X100 (a nonionic surfactant that has a hydrophilic polyethylene oxide chain and an aromatic hydrocarbon lipophilic or hydrophobic group), Transcutol-HP (a hydrophilic co-surfactant), were prepared [159,160]. The prepared systems were investigated for wound healing potential in rats in comparison with pure *Piper cubeba* oil and standard gentamycin. In this study was proposed a combination between *Piper cubeba* and gentamicin, the formulation prepared for synergic effects. The investigation proved that SNEDDS use presents significant enhancement in collagen content but no signs of inflammatory cells. The inhibition of the bacterial infection was attributed to the *Piper cubeba* EO contribution and it was considered the most efficient method to stimulate wound healing.

Poly(vinylidene fluoride) (PVDF) membranes with dual porosity were prepared through a technique mimicking polypore fungi obtained by non-solvent-induced phase separation [161]. The membranes were further charged with tea tree EO, to confer antimicrobial properties. The study recommended PVDF membranes, with easily tuned asymmetric channel-like porosity and controlled pore size, as ideal candidates for drug delivery applications. The new system presented the possibility to store controlled amount of bioactive component within the membrane porosity, and limited loss of the EO. The antimicrobial properties of PVDF membranes loaded with tea tree EO were tested against *Escherichia coli* and *Staphylococcus aureus*, as model microorganisms, and they showed antibacterial activity against both of them. In addition, the PVDF membranes embedding tea tree EO showed good cytocompatibility on mouse mesenchymal cells (C3HIOT1) and mouse myoblasts (C2C12).

A recent research study presented the encapsulation of *Saccocalyx satureioides* Coss. et Durieu EO into nanoemulsions based on 1% of Tween (a polyethylene sorbitol ester), by high-pressure homogenization [162]. It was found that the crystallization of the lipid droplets of the nanoemulsion by rapid cooling at the end of the high-pressure homogenization processing affected the volatile composition of the oil and of the prepared nanoemulsions. A positive effect on antioxidant and anticancer activities was registered, and, as a result, the technique was recommended for use in obtaining pharmaceuticals and in cancer treatment.

The anti-inflammatory and antalgic potency of the *Rosmarinus officinalis* L. EO, and of its nanoemulsion, prepared by using a low-energy loading methodology [163] in the presence of Tween 20, were investigated [164]. The *Rosmarinus officinalis* L. is known for its primarily content of limonene, camphor, and 1,8-cineole. The oil demonstrated a potentiated effect in the nanoemulsion formulation and a potent antalgic effectiveness in doses 600 times lower than those applied with EO alone. What was responsible for the anti-inflammatory and antalgic effects observed in the experimental results, with this new prepared nanoemulsion, was the camphor molecule, which presented the largest number of interactions with the therapeutic targets related to the inflammatory process.

Recently, a biobased compound, namely poly(ethylene brassylate-co-squaric acid), synthesized through the ethylene brassylate macrolactone ring-opening and copolymerization with squaric acid, was used for the encapsulation of thymol, a natural monoterpenoid phenol found in thyme oil, a compound with strong antiseptic properties [165]. The antimicrobial activity of poly(ethylene brassylate-co-squaric acid) thymol complex was investigated against eight different reference strains namely: bacterial strains—*Staphylococcus aureus* ATCC25923, *Escherichia coli* ATCC25922, *Enterococcus faecalis* ATCC 29212, *Klebsiella pneumoniae* ATCC 10031 and *Salmonella typhimurium* ATCC 14028, yeast strains represented by *Candida albicans* ATCC10231 and *Candida glabrata* ATCC 2001, and the fungal strain *Aspergillus brasiliensis* ATCC9642.

Even if they are not part of this special category of the polymers, we must also mention nanostructured lipid carriers as biocompatible and non-toxic systems. They are well known for the use in the pharmaceutical field due to their enhanced drug loading capacity and targeting efficiency with the possibility of site-specific delivery, drug release modulation flexibility, and improved stability [166,167]. Nanostructured lipid carriers (NLC) were tested for encapsulation of some EOs, namely *Rosmarinus officinalis* L., *Lavandula x intermedia* “Sumian”, *Origanum vulgare* subsp. hirtum and *Thymus capitatus* EOs, oils selected based on their antioxidant and anti-inflammatory activities [168,169]. The used NLC prepared from specific non-ionic surfactant systems were produced by phase inversion temperature (PIT) and high-pressure homogenization (HPH). According to the in vitro results, it was concluded on the enhancement of the biocompatibility and reduction of the cytotoxicity of the pure oils structured in the new complex state.

## 3. Machine Learning Analysis in Support of the EOs Use

The use of machine learning (ML) in the EOs investigation is carried out in several directions. This, in the context in which these products find more uses, and the diversity of their origins, their compositions, as well as the multiple uses as singular, complementary, and/or synergistic compounds, especially in the biomedical field, require a detailed and in-depth evaluation [170].

For example, Lucy Owen et al. article concluded on the importance of the quantitative structure activity relationship (QSAR) studies on antimicrobial effects of EOs components, which may provide promising targets for identification and future bacterial screening [171]. The authors investigated the structure–activity relationship of EOs compounds for target identification and optimization. In this context, 12 EO components were tested, and minimum inhibitory concentrations (MICs) of EO components were determined against *Escherichia coli* and *Staphylococcus aureus* using a microdilution method, which were further compared to those from the literature. MICs and FORGE software by consideration of electrostatic and steric parameters were used to generate a 3D qualitative SAR model. They found that the steric effects of some EO components have been shown as a structural commonality between inhibitory molecules, with the electrostatic properties and topography correlating well with the lowest MICs. Thus, between the investigated EOs carvacrol and menthol presented the lowest MICs against *Escherichia coli* and *Staphylococcus aureus*, with highly correlated electrostatic topographies.

Researchers used genetic algorithm and multiple linear regression (GA-MLR), partial least square (GA-PLS), kernel PLS (GA-KPLS), and the Levenberg Marquardt artificial neural network (L-M ANN) techniques in order to investigate the correlation between retention index (RI) and descriptors for 116 diverse compounds in EOs of six Stachys species [172]. By using the mentioned techniques, it was found that RI of EOs possesses some nonlinear characteristics. At the same time, it was confirmed the possibility to apply L-M ANN model to describe the molecular structure characteristic of these compounds. In this way, the artificial intelligence introduced in chromatographic treatment allows solving the automatic compensation of random retention times [173].

In another investigation random forests machine learning algorithm was applied to the classification of 20 different EOs in order to use their differences in chemical profiles as chemical markers for EO classification and determination of the quality [174]. The total chromatogram average mass spectra (TCAMS) and segment average mass spectra (SAMS) were created from three-way raw gas chromatography–mass spectra data, and the resulted SAMS data set showed superior potential for quality assurance, compared with TCAMS, while TCAMS is much faster and more readily created. Moreover, both models showed excellent performance in the classification of EO classes.

In a machine learning algorithm application, the development of quantitative activity–composition relationship classification models was used to point out the EO chemical components more involved in the inhibition of biofilm production [175]. The authors concluded the performed classification model, validated by five performance metrics, as an example, showing machine learning as a tool to investigate complex chemical mixtures, and possibly in prospective experiments, understanding the mechanism by which EOs act. Moreover, in case of the increased number of experiments, the generated model enables the identification of EO blends, specifically designed to obtain products with strong anti-biofilm efficacy, applicable in many fields: airborne decontamination, products for dermatological and respiratory tract infections, etc.

Due to EO use as a natural antibiotic in complementary and alternative medicine, artificial neural network (ANN) was applied for the prediction of some EO antimicrobial activity [176]. In this context, 49 EOs, extracts, and/or fractions with chemical composition and antimicrobial activity extracted from the National Committee for Clinical Laboratory Standards compliant works (NCCLS) were used. The output data reflecting the antimicrobial activity of the tested EOs against four common pathogens, namely *Staphylococcus aureus, Escherichia coli, Candida albicans*, and *Clostridium perfringens* as measured by standardized disk diffusion assays, were implemented in the fast artificial neural networks software. The artificial neural networks were able to predict > 70% of the antimicrobial activities within a 10 mm maximum error range, and two or three different bioactivities were predicted at the same time.

Based on the experimental data and due to the interest in the development of new approaches in the prevention and treatment of bacterial biofilm under the influence of EOs, which can modulate biofilm production, with unpredictable results leading to either bacterial biofilm inhibition or reduction, depending on EO chemical composition and on type of microorganism, a machine learning application for the complex matrix of data from 89 EO chemical analyses was investigated [177].

In another investigation study that used the computational models, a computational tool was developed to select EOs that have antimicrobial activities in relation to their effective compounds, but without complex laboratory analysis [178]. The authors underlined and justified their study, which can save money and time and enhance consistency of final products. Thus, they established a new computational tool named the Essential Oil Reduction and Optimization Tool (EOROT) to determine the best and optimal EO in inhibiting the activity of the bacteria growth. The main phases of the study consisted in determining the EO chemical composition, reducing the number of the chemical compounds by specifying the main antibacterial compounds and establishing the rules of the relationship between these compounds, and finally, deciding on the EOs with high antibacterial activities, and concluding for the most-suited EO for every type of the bacteria. Unfortunately this research was limited to only 24 EOs tested toward 17 kinds of bacteria.

As Noha E. El-Attar’s article mentioned, the bioactivity of EOs as dynamic blends depends and varies according to their chemical constitution and structure [179]. The research team proposed the development of machine learning-based computational models, namely Multiclass Neural Network (MNN), and Convolutional Neural Network (CNN), as deep learning techniques, to categorize and predict the biological activities of EOs based on their chemical construction variations, without recourse to the in vitro experiments. Data and comparison between these algorithms showed that CNN outperformed MNN with an accuracy rate of about 98.13%, while MNN recorded 81.88%.

## 4. Opportunities, Challenges, and Prospects

The multiple benefits offered by EOs, designed by their complex nature, ensure a general growth in terms of both their use, and the areas in which these generous products can find their applicability, in a unique, synergistic, and/or complementary way. At the same time, the nature of EOs, which can promise antimicrobial, antioxidant, antitumor, antifungal, and anti-inflammatory effects, must be protected from environmental conditions, such as light, heat, and humidity, in order to be fully effective in the areas in which they are used. Thus, one of the main challenges faced by scientific researchers is to find structures that ensure the appropriate matrices for encapsulating EOs and their protection, as well as their controlled release under predetermined conditions. In this context, synthetic polymers, together with natural ones, can be an inexhaustible source of networks/matrices for the encapsulation of EOs to ensure their protection. In this regard, researches can generate innovative structures. Moreover, due to the fact that EOs have no major side effects, as most of the conventional medicines and drugs, they can be used to help treat numerous health problems, such as cardiovascular problems, Alzheimer’s, bronchitis, and aromatherapy, for example. Another requirement for researchers is the need to create more demand for beneficial EOs, by seeking new EO biological activities, and investigating the pharmaceutical possibilities of EOs; thus, creating the basis for more possible applications in pharmaceutical and medical applications, or even adjuvants to current medications.

A concise presentation of the natural or synthetic polymeric structures used for the encapsulation of EOs, to protect them from environmental conditions, but also for their controlled release, the loading method, the biological activities provided for the systems made, and the bibliographic references used, are provided in the Appendix A to this review.

## Figures and Tables

**Figure 2 pharmaceutics-13-00631-f002:**
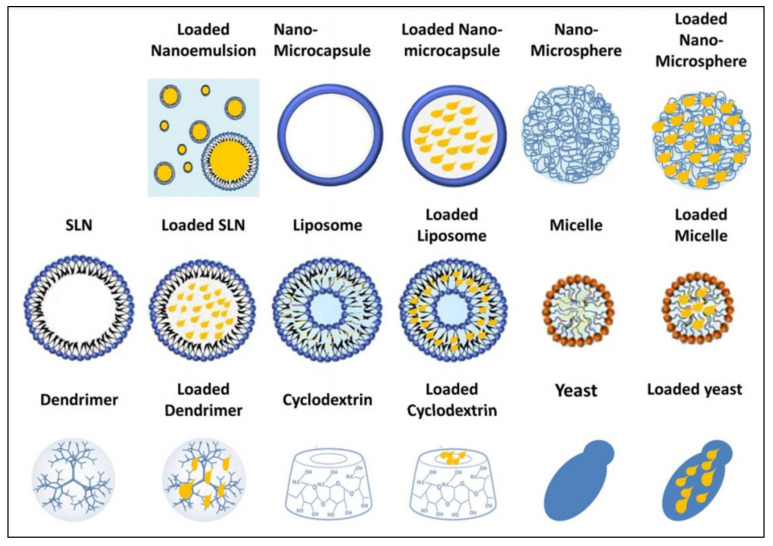
Various structures for bioactive oils encapsulation. Reprinted with permission from ref. [46]. Copyright 2016, Elsevier Ltd.

**Figure 3 pharmaceutics-13-00631-f003:**
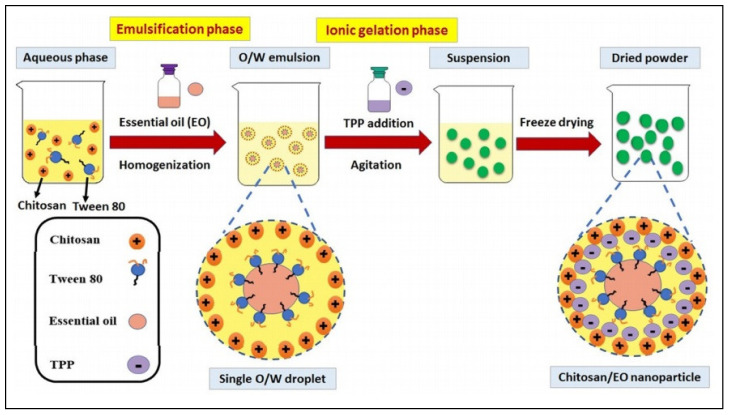
Schematic illustration of emulsification/ionic gelation procedure. Reprinted with permission from ref. [108]. Copyright 2019, Elsevier Ltd.

**Figure 4 pharmaceutics-13-00631-f004:**
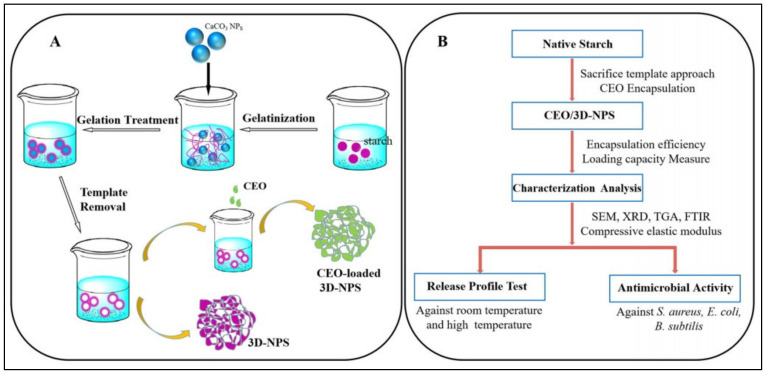
Illustration of the 3D-nanoparticles formation for the clove EO encapsulation (**A**), and the experimental flowchart (**B**). Reprinted with permission from ref. [132]. Copyright 2019, Elsevier Ltd.

**Figure 5 pharmaceutics-13-00631-f005:**
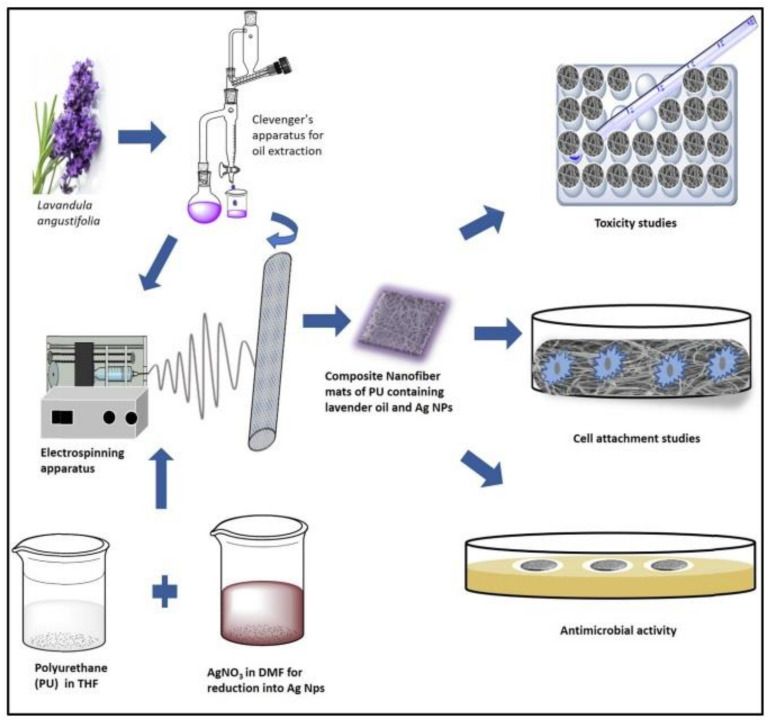
Scheme illustrating the nanofiber composites fabrication through electrospinning, from polyurethane encasing lavender oil and Ag nanoparticles. Reprinted with permission from ref. [146]. Copyright 2019, Elsevier Ltd.

## Data Availability

Not applicable.

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
