# Peer review of "Polymeric Carriers Designed for Encapsulation of Essential Oils with Biological Activity"

_pharmaceutics, 2021, doi:10.3390/pharmaceutics13050631_

Round 1
Reviewer 1 Report
The review paper is written in a correct way. It contains important information on the materials, technologies, and methods used to encapsulate essential oils.
Author Response
Answers to the Reviewers
The manuscript, entitled:
Polymeric carriers designed for encapsulation of essential oils with biological activity
authors:
Aurica P. Chiriac, Alina G. Rusu, Loredana E. Nita, Vlad M. Chiriac, Iordana Neamtu, Alina Sandu
Firstly, we thank the Reviewers by helping us to improve our paper. The corrections were made and the manuscript was re-written accordingly with indications.
Comments from reviewer:
- The review paper is written in a correct way.It contains important information on the materials, technologies, and methods used to encapsulate essential oils.
Response to the reviewer’s comments (1)
Thank you for your appreciations.
Reviewer 2 Report
Paper entitled "Polymeric carriers designed for encapsulation of essential oils with biological activity" focuses on different polymers of natural and synthetic origin used for encapsulation of essential oils.
Although it is obvious that authors invested lots of time and effort in collecting huge amount of data, the presentation of data (the paper itself) is not satisfactory. Manuscript if characterised with long sentences and lots of grammatical errors which makes manuscript very difficult to read, and sometimes understand.
Apart from mentioned, it seems that authors lost the focus through the paper. The topic of the paper suggest that the aim was to discuss the advantages and disadvantages of different polymers, while in the paper itself lots of lines are related to biological activity of EOs.
Author contribution in terms of data interpretation is missing. Without, or with rare author comments and just listing data the paper itself resembles giant patchwork. Chapter 2.8. Machine learning analysis in support of the EOs use - does not seem to be part of the paper.
I suggest major revision with text rewriting and focusing on pros (active substance release, improved pharmacokinetics, protective effect…) and cons of different polymers. Extensive language editing is strongly recommended.
Author Response
Answers to the Reviewers
The manuscript, entitled:
Polymeric carriers designed for encapsulation of essential oils with biological activity
authors:
Aurica P. Chiriac, Alina G. Rusu, Loredana E. Nita, Vlad M. Chiriac, Iordana Neamtu, Alina Sandu
Firstly, we thank the Reviewers by helping us to improve our paper. The corrections were made and the manuscript was re-written accordingly with indications.
Comments from reviewer:
Paper entitled "Polymeric carriers designed for encapsulation of essential oils with biological activity" focuses on different polymers of natural and synthetic origin used for encapsulation of essential oils.
Although it is obvious that authors invested lots of time and effort in collecting huge amount of data, the presentation of data (the paper itself) is not satisfactory. Manuscript if characterised with long sentences and lots of grammatical errors which makes manuscript very difficult to read, and sometimes understand.
Apart from mentioned, it seems that authors lost the focus through the paper. The topic of the paper suggest that the aim was to discuss the advantages and disadvantages of different polymers, while in the paper itself lots of lines are related to biological activity of EOs.
Author contribution in terms of data interpretation is missing. Without, or with rare author comments and just listing data the paper itself resembles giant patchwork. Chapter 2.8. Machine learning analysis in support of the EOs use - does not seem to be part of the paper.
I suggest major revision with text rewriting and focusing on pros (active substance release, improved pharmacokinetics, protective effect…) and cons of different polymers. Extensive language editing is strongly recommended.
Response to the reviewer’s comments (2)
- Although it is obvious that authors invested lots of time and effort in collecting huge amount of data, the presentation of data (the paper itself) is not satisfactory. Manuscript if characterised with long sentences and lots of grammatical errors which makes manuscript very difficult to read, and sometimes understand.
- According to the reviewers' instructions, the article was checked, grammatical errors removed, and some phrases and sentences were rewritten
- Also, each paragraph, sentence and sentence was checked and we followed the re-phrasing of some of them for improved coherence.
- Apart from mentioned, it seems that authors lost the focus through the paper. The topic of the paper suggest that the aim was to discuss the advantages and disadvantages of different polymers, while in the paper itself lots of lines are related to biological activity of EOs.
As it is mentioned including in abstract the purpose of the review is at follows:
The article reviews the possibilities of encapsulating EOs, due to their multiple benefits, in order to protect them from the environmental conditions but also for their controlled release. Thus, there are presented the natural polymers and also, the synthetic macromolecular chains that are commonly used as networks for embedding EOs owing to their biodegradability and biocompatibility, in interdependence with the encapsulation methods and with the potential applicability of the bioactive blend structures.
At the same time it is not the role of this review to find disadvantages in the use of polymeric matrices for coupling essential oils but only to highlight the possibilities of encapsulating them in order to preserve the properties of the oils, challenges and prospects in this context being presented in the conclusions of the article.
- Author contribution in terms of data interpretation is missing.
Chapter 2.8. Machine learning analysis in support of the EOs use - does not seem to be part of the paper.
The intention of the authors was precisely a review of the latest achievements in the field of encapsulation of essential oils in polymer matrices, respectively the applicability of the prepared bioactive structures.
Chapter 2.8, namely machine learning analysis in support of the EOs use presents and underlines the possibilities of using artificial intelligence to evaluate the bioactivity of EOs in direct correlation with their chemical constitution and structure, in order to avoid complex laboratory analysis, for saving money and time, and for enhancement of the final consistency of the products. The use of artificial intelligence in this field can only be an opportunity not to be missed, and it is a challenge as it represents beneficial solutions in the domain of essential oils applicability and use.

Reviewer 3 Report
This interesting review “Polymeric carriers designed for encapsulation of essential oils 2 with biological activity” addresses an overview of research studies reporting the delivery of essential oils in polymeric carriers to protect them from the environmental conditions and control their release. Thus, the natural synthetic polymers, their biodegradability and biocompatibility, the preparation methods and the potential applicability of formulations are presented. The subject is very interesting, the data are well presented, and the previously studies are well described. I suggest the publication in “Pharmaceutics” after minor revisions.
In the Introduction, lines 22-28, the sentence is not clear and must be rephrased and the concept well explained, especially the sentence (lines 23-24) …by the need to address the issue in specific fields through compounds and natural structures.
In the Introduction, line 37, the word “inexpensiveness” is inappropriate because the essential oils are very expensive.
In the Introduction, lines 37-38, the sentence “and also, to a synergistic therapeutic effect with the prescribed medical treatment” in this context is not clear should be clarified.
In the Introduction, line 40, please check the letter “o” of “Eos”.
In the Introduction, lines 38-41, and 49-51, in these sentences, the abbreviation “Eos” has been reported 3 times in each sentence.
In the whole manuscript, lines 49, 199, 228, the letter “s” of “EOs” before encapsulation must be eliminated.
In the Introduction, line 59, the letter “s” of “EOs” before mechanism must be eliminated.
In the Introduction, line148, the letter “s” of “EOs” before potential must be eliminated.
Author Response
Answers to the Reviewers
The manuscript, entitled:
Polymeric carriers designed for encapsulation of essential oils with biological activity
authors:
Aurica P. Chiriac, Alina G. Rusu, Loredana E. Nita, Vlad M. Chiriac, Iordana Neamtu, Alina Sandu
Firstly, we thank the Reviewers by helping us to improve our paper. The corrections were made and the manuscript was re-written accordingly with indications.
Comments from reviewer:
This interesting review “Polymeric carriers designed for encapsulation of essential oils with biological activity” addresses an overview of research studies reporting the delivery of essential oils in polymeric carriers to protect them from the environmental conditions and control their release. Thus, the natural synthetic polymers, their biodegradability and biocompatibility, the preparation methods and the potential applicability of formulations are presented. The subject is very interesting, the data are well presented, and the previously studies are well described. I suggest the publication in “Pharmaceutics” after minor revisions.
In the Introduction, lines 22-28, the sentence is not clear and must be rephrased and the concept well explained, especially the sentence (lines 23-24) …by the need to address the issue in specific fields through compounds and natural structures.
In the Introduction, line 37, the word “inexpensiveness” is inappropriate because the essential oils are very expensive.
In the Introduction, lines 37-38, the sentence “and also, to a synergistic therapeutic effect with the prescribed medical treatment” in this context is not clear should be clarified.
In the Introduction, line 40, please check the letter “o” of “Eos”.
In the Introduction, lines 38-41, and 49-51, in these sentences, the abbreviation “Eos” has been reported 3 times in each sentence.
In the whole manuscript, lines 49, 199, 228, the letter “s” of “EOs” before encapsulation must be eliminated.
In the Introduction, line 59, the letter “s” of “EOs” before mechanism must be eliminated.
In the Introduction, line148, the letter “s” of “EOs” before potential must be eliminated.
Response to the reviewer’s comments (3)
- In the Introduction, lines 22-28, the sentence is not clear and must be rephrased and the concept well explained, especially the sentence (lines 23-24) …by the need to address the issue in specific fields through compounds and natural structures.
- The sentences were re-phrased as follows:
- The interest for the current use of essential oils (EOs) is motivated by raising awareness of health and well-being among consumers, and it is confirmed through the large number of dedicated scientific articles, which is also sustained by the EOs market that is significantly driven by the growing demand for natural products, which usually have less or no alteration. At the same time, these aspects are also supported by the need to address the issue in specific fields through compounds and natural structures.
- In the Introduction, line 37, the word “inexpensiveness” is inappropriate because the essential oils are very expensive.
- Inexpensiveness was replaced with availability
- In the Introduction, lines 37-38, the sentence “and also, to a synergistic therapeutic effect with the prescribed medical treatment” in this context is not clear should be clarified.
The sentence was completed as follows:
EOs are suitable as complementary medicine treatments owing to their pleasantness and availability, and also, to a synergistic therapeutic effect with the prescribed medical treatment, as for example the antibiotics.
- In the Introduction, line 40, please check the letter “o” of “Eos”.
- It was corrected.
- In the Introduction, lines 38-41, and 49-51, in these sentences, the abbreviation “Eos” has been reported 3 times in each sentence.
The sentence was re-written as follows:
Aspects related to the significant antimicrobial potential against multidrug-resistant pathogens of the EOs, their synergetic effect when more oils are mixed [6-8], or synergistic activity when used in combination with known drugs [9-14], are also presented in a recent review.
and
A particularly important direction is represented by the problem of EOs encapsulation, as well as exploiting the synergism between oils constituents and medicines to produce cooperation and/or a combined effect.
- In the whole manuscript, lines 49, 199, 228, the letter “s” of “EOs” before encapsulation must be eliminated.
- Lines 49, 199, 228, the sentences were corrected.
- In the Introduction, line 59, the letter “s” of “EOs” before mechanism must be eliminated.
The sentences was corrected.
- In the Introduction, line148, the letter “s” of “EOs” before potential must be eliminated.
The sentences was corrected.

Round 2
Reviewer 2 Report
The Authors have improved the manuscript according to suggestions, or provided reasonable explanations if not doing so.
I agree for the paper to be accepted in present form.